# ACTIVE PROCEDURE PLANNING WITH UNCERTAINTY-AWARENESS IN INSTRUCTIONAL VIDEOS

## ABSTRACT

Procedure planning involves the generation of a sequence of steps that bring a specific start state to the desired goal state. Both states are given as visual observations in the case of planning from instructional videos. This is a challenging task due to ambiguities in the visual representations of states and variations arising from multiple feasible plans. Existing approaches address these challenges by adopting strong visual representation learning methods and sophisticated reasoning mechanisms. However, the decision process is passive in the sense that both the visual observations and the reasoning process are fixed during the planning phase. In this paper, we propose an active procedure planning approach that takes account of uncertainties arising from imperfect visual observations and task plan variations. In particular, we develop quantitative metrics to evaluate task uncertainty and use them to guide the selection of additional visual observations. Empirical results show that visual observations driven by uncertainty-awareness lead to significantly higher performance gain compared to opportunistic visual observations. The findings are useful for developing trusted and explainable AI models for procedure planning. The code will be released upon paper acceptance.

## 1 INTRODUCTION

Procedure planning in instructional videos is concerned with the generation of action plans given the visual observations of a start state ($o_s$) and a goal state ($o_g$) (Chang et al., 2020; Sun et al., 2021). This task is of practical relevance because learning and reasoning are grounded on real-world scenes, often with the presence of human actors interacting with the environment. It is different from and perhaps more challenging than procedure planning in the natural language domain (Wei et al., 2021), or in simulated environments (Shridhar et al., 2019). Indeed, it represents a proxy of the anticipated future scenario where an agent co-exists with a human and provides in situ assistance, *e.g.*, helping a person prepare a recipe. The ability to observe the world state and make action plans with or for humans is apparently necessary in such situations.

Current approaches for procedure planning in instructional videos adopt fixed conditions when generating action plans. By fixed conditions, we mean that the input to the decision module, usually defined as a tuple $(o_s, o_g)$, is kept constant throughout the planning process. In other words, there is no information exchange between the agent and the external world once planning starts, so that it maintains a fixed internal representation of the task and world state. Consequently, the agent is not trained to interact with the environment and gather useful information for decision-making.

We aim to equip the agent with the ability to seek additional information based on an educated judgment of the world states. In the current problem context, uncertainty arises from ambiguities with respect to the task conditions. In particular, procedure planning is conditioned on visual observations (video clips) of the start and goal states, where a stack of image frames is arbitrarily extracted from a video to represent a state. However, the extracted images may not clearly denote the action steps or the task. Furthermore, there are immense variations in procedure plans. As illustrated in Fig. 1 (Left), three videos showing the task *make meringue* share the same start state (*pour egg*) and goal state (*whisk mixture*). However, the intermediate states are different, representing three feasible trajectories. It is extremely difficult to converge to a single path purely from the semantic meaning of the start and goal states. A quick remedy to this problem is to make additional observations to disambiguate the states and constrain the trajectories. As shown in the right pane of Fig. 1, given

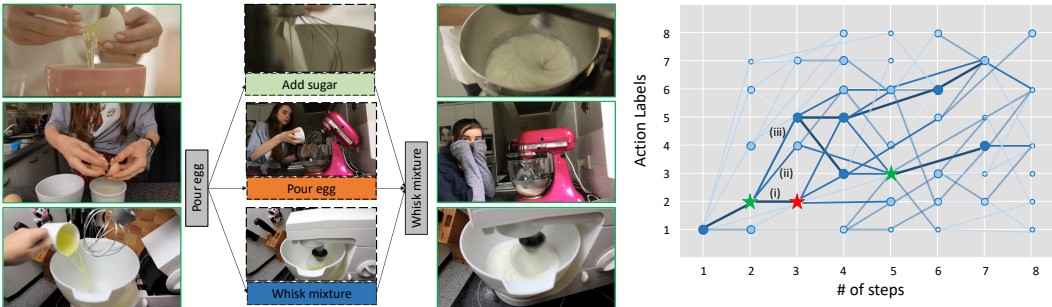

Figure 1: Ambiguity in task planning. Left: *Make Meringue*: there are multiple feasible trajectories given similar start state (*pour egg*) and goal state (*whisk mixture*). Right: Expert trajectories of a long horizon task (adapted from (Bi et al., 2021). A darker color indicates nodes/paths that are visited more frequently. In a conceived planning task, Green stars denote start and goal states. A red star denotes an observation that results in a recognized intermediate state. It helps to prune alternative routes (ii)&(iii) that are otherwise plausible.

the start and goal states (denoted as green stars), adding an observation (denoted as red star) can help the agent effectively remove alternative routes, provided that the perceptual module can correctly discern the action. Meanwhile, each observation incurs costs related to data acquisition and increased cycle time. Therefore, it is worthwhile to explore when and how to apply additional visual observations to achieve anticipated performance gain, while keeping the cost low.

To tackle this issue, this study investigates the conditions under which additional visual observations result in beneficial outcomes in terms of planning accuracy and cost efficiency. We posit that visual observations are more useful when higher uncertainty is involved in the prediction outcome. Based on this assumption, it is crucial to estimate uncertainty in procedure planning for individual task instances. We design comprehensive uncertainty metrics that combine a calibrated confidence score and a task sequence variation score. Furthermore, we propose an active planning approach that selectively adds visual observations based on the estimated uncertainty. We incorporate the active planning mechanism into the latest diffusion-based procedure planning models (Wang et al., 2023; Fang et al., 2023), and evaluate the effectiveness of the method on CrossTask (Zhukov et al., 2019) and COIN (Tang et al., 2019) datasets. It is shown that our method consistently boosts prediction accuracy on both datasets and for different planning horizons. We find that the calibrated confidence score captures planning uncertainty more effectively than the task variation score.

**Contributions**. (1) We propose a new approach of active procedure planning that enables flexible visual observations to disambiguate the planning task. (2) We analyze the source of uncertainty and develop comprehensive metrics to evaluate the uncertainty of procedure planning. (3) Through extensive experiments, we find new empirical evidence regarding the effect of active procedure planning on the accuracy of generated plans. Our method represents a paradigm shift in procedure planning from passive reasoning to active/adaptive learning and reasoning. It paves the way toward equipping AI agents with the ability to interact with the environment in seek of useful information to solve problems. Importantly, the ability to reason about uncertainty related to the perceptual inputs and decisions is a critical element of trusted and explainable AI (Kok & Soh, 2020).

## 2 RELATED WORK

### 2.1 PROBABILISTIC PROCEDURE PLANNING

A straightforward way to handle uncertainty related to task variations is to build reasoning models that capture the statistical distribution of the data. Bi *et al.* (2021) proposed an exterior-model generative adversarial imitation learning (Ext-GAIL) that implements Bayesian inference to deal with the uncertainty of the environment. Zhao *et al.* (2022) built a generative module trained with generative loss that can produce multiple feasible plans at inference time. Recently, Wang *et al.* (2023) formulated procedure planning as a conditional sampling process and proposes a projected diffusion model to account for the probabilistic planning process. Fang *et al.* (2023) extended the diffusion-based

model with a hierarchical reasoning mechanism to further boost the performance. However, all the above methods consider uncertainty as an inherent characteristic of data distribution and develop techniques to simulate the distribution. They do not account for the possible information deficiency due to the ambiguity of inputs and conditions. Therefore, there is no mechanism to update the internal representation during the reasoning process.

To make the planning decisions grounded on perceptual semantics, early works leverage the conjugate relationship between actions and states to provide constraints on the learning and reasoning process (Chang et al., 2020; Sun et al., 2021; Bi et al., 2021). They generate procedure plans in an autoregressive manner to harness the inter-dependency of actions and states. However, information exchange happens only between sub-modules of the system, which does not help if the initial internal representation is inaccurate or incomplete. Besides, such a mechanism leads to error propagation, and in turn performance deterioration, especially for long sequences (Zhao et al., 2022). Recent works resort to external knowledge to enhance the reasoning process. For example, Zhao *et al.* (2022) incorporated a learnable global memory module to augment the plan generation; Patel *et al.* (2023) leveraged the reasoning power of pre-trained language models to enhance sequence planning; Lu *et al.* (2022) built a procedure knowledge graph trained on both text dataset and video corpus to provide additional supervision to the downstream planning tasks. However, such external knowledge does not address the information deficiency of situated planning tasks. The issue cannot be eradicated by stronger visual representation learning models (Lin et al., 2022; Xu et al., 2021; Zhao et al., 2023) either.

## 2.2 ACTIVE LEARNING AND UNCERTAINTY EVALUATION

The idea of active learning stems from the need to train machine learning models with fewer labeled data due to high annotation costs. In the era of deep neural networks (DNNs), many deep active learning approaches have been proposed that focus on how to choose representative data points based on knowledge of data distribution (Sener & Savarese, 2017; Gal et al., 2017; Liu et al., 2021; Ren et al., 2020). It has led to notable development in few-shot learning (Woodward & Finn, 2017; Vinyals et al., 2016) and meta-learning (Ravi & Larochelle, 2018) for various applications, such as image recognition (Lee et al., 2019), navigation (Chen et al., 2019), object detection (Yuan et al., 2021), and action recognition (Roitberg et al., 2021; Subedar et al., 2018). Nevertheless, this notion is different from active procedure planning, which is concerned with selectively acquiring perceptual inputs to reduce scene ambiguity/uncertainty.

A closely related field of research is uncertainty evaluation and model calibration. It is observed that modern DNNs, while being more accurate, are often miscalibrated, *i.e.*, the reported confidence level of a prediction does not reflect the true correctness likelihood (Guo et al., 2017; Minderer et al., 2021; Nixon et al., 2019). Many approaches have been proposed to quantify uncertainty, such as prior networks (Malinin & Gales, 2018), evidential neural networks (Sensoy et al., 2018), stochastic variational inference (Blundell et al., 2015), ensemble methods (Wen et al., 2020), *etc*. Interested readers may refer to (Abdar et al., 2020; Gawlikowski et al., 2021; Mena et al., 2021) for reviews of uncertainty estimation in modern deep neural networks. This research will evaluate the uncertainty of procedure planning as a means to guide the data acquisition process. We do not intend to develop new methodologies to evaluate uncertainty *per se*.

Embodied AI is another relevant field of research (Lu et al., 2022; Zhang et al., 2022; Zheng et al., 2022; Inan et al., 2023). However, the interactive process specified in this study is different from planning tasks in embodied AI, where an agent explores a space and gets feedback signals in a virtual environment. The steps in individual instructional videos are fixed and the interaction is restricted to extra observations of the next steps to determine the entire task process. More importantly, visual learning and reasoning in real-world videos are more challenging than in simulated environments.

## 3 PROBLEM FORMULATION

Procedure planning in instructional videos is traditionally formulated as a conditioned sequence generation problem (Chang et al., 2020), namely given the visual observations of a start state ($o_s$) and a goal state ($o_g$), a model is tasked to produce a plan in the form of a sequence of actions $a_{1:T}$ that, when executed, will facilitate the transition from $o_s$ to $o_g$ in $T$ steps, where $T$ is called the

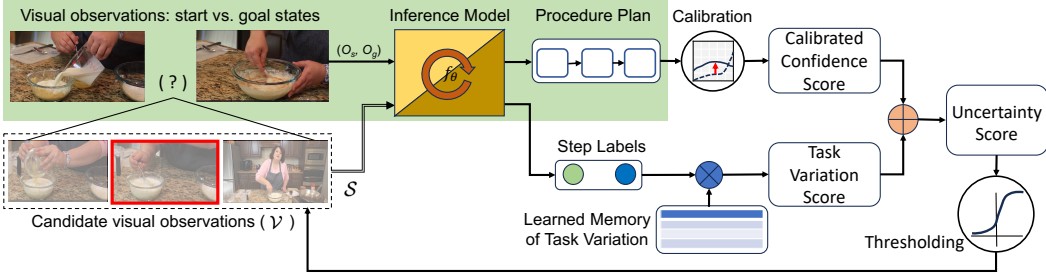

Figure 2: Overview of active procedure planning with uncertainty-awareness. The box shaded in green denotes the traditional procedure planning process.

planning horizon. With active procedure planning, the agent is allowed to perform intermediate visual observations during inference time so as to resolve potential ambiguities. The entire set of intermediate states (based on visual representations) is denoted as $\mathcal{V}$, from which a subset of additional visual observations $\mathcal{S}(\subseteq \mathcal{V})$ can be drawn. When $\mathcal{S} = \Phi$, *i.e.*, no additional visual observation is made, the problem is identical to conventional passive procedure planning. Assuming that every additional visual observation incurs a cost, the problem is reduced to how to select a subset $\mathcal{S}$ that boosts planning accuracy while keeping the cost within a pre-defined budget.

## 4 APPROACH

Active procedure planning extends legacy procedure planning with uncertainty-based sampling. It is carried out during the inference phase and does not affect the training of the plan generation model. As illustrated in Fig. 2, a legacy procedure planning method (box shaded in green) generates procedure plans using a trained inference model[1]. To improve the prediction accuracy, we propose to add new visual observations when needed. The key insight is that additional visual observations are most useful when the inference model is unable to produce valid procedure plans. However, there is no external feedback (*e.g.*, human-in-the-loop) on the correctness of the generated plans. Hence, it is crucial to get a judicious uncertainty estimation based on the model's self-awareness of its prediction.

### 4.1 UNCERTAINTY ESTIMATION

This research considers two aspects of uncertainty, namely (1) uncertainty arising from task sequence variation, and (2) uncertainty related to the model's prediction confidence.

#### 4.1.1 TASK SEQUENCE VARIATION SCORE

As aforementioned, there can be multiple feasible trajectories given the start and goal states. It is postulated that the more trajectory variants, the higher the uncertainty involved. First, we analyze the characteristics of task trajectory distribution from training data. Assuming that data in the test set and training set is independent and identically distributed, we can use the distribution of sequence variants of the training set as a reference to infer that of the test set. In this sense, task sequence variation is considered as the prior knowledge of the data distribution. We estimate uncertainty related to task sequence variation as follows.

During training, given a pair of start and goal states $(o_s, o_g)$ extracted from a video clip, the ground-truth action sequence $a_{1:T}$ is available, where $a_1$ and $a_T$ correspond to the action labels of $o_s$ and $o_g$, respectively. In reality, for each tuple $(a_1, a_T)$, there could be multiple video instances that follow different trajectories, as illustrated in Fig. 3. For each video instance that corresponds to a trajectory, we register a visit to that path. Based on the number of visits to each path, we calculate the percentage of visits along individual paths. Let $M$ be the total number of distinctive trajectories, and

---

[1]For purpose of conciseness, we use the box "inference model" to denote the entire inference process of procedure planning, while omitting the technical details of the reasoning process.

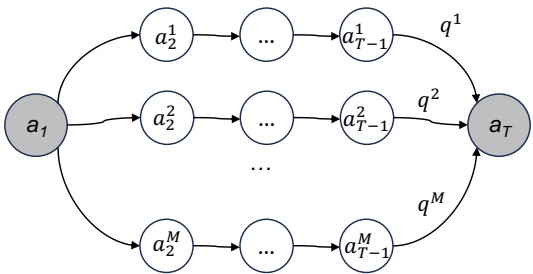

Figure 3: Uncertainty estimation based on task sequence variation.

$q^m (m = 1, ..., M)$ be the proportion of visits to path $m$ among all the visits. We use the Shannon entropy (Borda, 2011) of path visits as an indicator of uncertainty related to task sequence variation.

$$u_v(a_1, a_T) = -\frac{1}{||u_{v(T)}||} \sum_{m=1}^{M} q^m \log q^m. \tag{1}$$

where $||u_{v(T)}||$ is the maximum value for all $(a_1, a_T)$ tuples with planning horizon $T$. It serves to scale the uncertainty score of an individual $(a_1, a_T)$ tuple to be in the range of $[0, 1]$. Eq. 1 reflects the fact that higher uncertainty is involved when there are more trajectories and when data samples are evenly distributed among multiple trajectories. For example, if there is a single path, the uncertainty is effectively '0'; two paths with a $(50\%, 50\%)$ distribution of visits involves higher uncertainty than those with a $(10\%, 90\%)$ distribution. More details on the characteristics of task variation score are given in Appendix A.6.

Based on the entire training set, we obtain a learned memory of task variation that encodes the uncertainty for any valid $(a_1, a_T)$ tuple. During inference, the inference model generates the action plans $\hat{a}_{1:T} \sim f_\theta(o_s, o_g)$, where $f_\theta$ is a mapping function with learnable parameters $\theta$, $\hat{a}_1$ and $\hat{a}_T$ correspond to the predicted action labels of $o_s$ and $o_g$, respectively. The model can then retrieve the uncertainty score from the learned memory of task variation distribution based on the matching of $(\hat{a}_1, \hat{a}_T)$ versus $(a_1, a_T)$.

### 4.1.2 CALIBRATED PREDICTION CONFIDENCE SCORE

The action plan $\hat{a}_{1:T}$ generated by the inference model is always affixed with a predicted confidence score, which for the case of procedure planning, is the probability value computed from a softmax function in the last network layer. Although the confidence score can be interpreted as a prediction of uncertainty, is widely discussed that DNNs tend to be over-confident with the prediction, *i.e.*, the score is poorly calibrated. To ensure robust prediction of uncertainty, we adopt temperature scaling (Guo et al., 2017) to calibrate the confidence score. Specifically, for an individual step/action in a procedure plan, the calibrated confidence score is calculated as

$$\bar{p}_i = \max_k \sigma(z_i/\mathcal{T})^{(k)}, \tag{2}$$

where $i \in \{1, 2, ..., T\}$ is the index of an action in the planning horizon; $k$ is the total number of action label types; $\sigma$ denotes the sigmoid function; $z_i$ denotes the logit vector produced before the softmax layer; $\mathcal{T}$ is the temperature. An uncertainty score for the entire action sequence is computed based on the predicted confidence of individual steps:

$$u_c(a_1, a_T) = 1 - \min_i(\bar{p}_i). \tag{3}$$

The final uncertainty score is computed as the weighted uncertainties related to task variation and calibrated confidence. Let $0 \leq w_1, w_2 \leq 1$ (where $w_1 + w_2 = 1$) be the weights assigned to the task variation score and calibrated confidence score, respectively. The combined uncertainty score is computed as

$$u = w_1 u_v + w_2 u_c. \tag{4}$$

## 4.2 UNCERTAINTY-AWARE PROCEDURE PLANNING

Decisions on the selection of additional visual observations are made based on the uncertainty score (Eq. 4) by applying a threshold $\tau$, *i.e.*, if $u \geq \tau$, an additional observation is required. For tasks with planning horizon $T = 3$, there is only one candidate intermediate visual observation, which is selected as the input to the inference model. For longer planning horizons (*i.e.*, $T \geq 4$), one may choose to add one or a few additional observations; and for the latter situation, one may add them all at once or incrementally based on the quality of the generated procedure plan. In this study, we restrict the number of allowed additional observations to one without iterative observations, so as to control the cost of data acquisition. As shown in the experimental results, such a strategy is effective in improving performance. In addition, to choose one additional observation from multiple candidates, we adopt a simple strategy which is to pick the one near the temporal center of the video clip. Specifically, when $T$ is an odd number (*e.g.*, 3 and 5), we select the center observation; when $T$ is an even number (*e.g.*, 4 and 6), we opt for the center-right observation. We leave it as future work on more comprehensive strategies for selecting additional observations.

## 4.3 IMPLEMENTATION DETAILS

All experiments are performed using PyTorch (Paszke et al., 2019) on a machine with 4 NVIDIA RTX A5000 GPUs, and we employ the Adam optimizer (Kingma & Ba, 2015) for training. The learning rate for Adam is adjusted based on the specific dataset. Detailed hyper-parameter settings for each dataset are given in Appendix A.5.

## 5 EXPERIMENTS

### 5.1 EVALUATION PROTOCOL

**Datasets.** Our method is evaluated on two datasets: CrossTask (Zhukov et al., 2019) and COIN (Tang et al., 2019). CrossTask (Zhukov et al., 2019) has 2,750 instructional videos, collected for 18 different tasks with a total of 133 actions and an average of 7.6 actions in each video. In this dataset, several common actions, such as *pour water*, *pour milk*, and *stir mixture*, are shared across multiple tasks. COIN (Tang et al., 2019) is a large-scale dataset, with 11,827 instructional videos, 180 different tasks, and 778 actions, with each action exclusively associated with a single task. For both datasets, visual features are extracted from encoders trained on the HowTo100M dataset (Miech et al., 2019).

**Metrics.** Following previous works (Sun et al., 2021; Zhao et al., 2022; Wang et al., 2023), we adopt three metrics to evaluate the performance of our method. (i) Success Rate (*SR*) defines a plan as successful only when every action precisely matches the ground-truth sequence. (ii) Mean Accuracy (*mAcc*) computes the average accuracy of actions at each individual time step, where a predicted action is deemed correct if it matches the corresponding action in the ground truth at that specific time step. (iii) Mean Intersection over Union (*mIoU*) considers the predicted and ground-truth action sequences as sets, and computes the intersection between these sets. Hence, *mIoU* is indifferent to the action order and solely signifies whether the model correctly captures the essential set of steps required to execute the plan. Different from traditional passive procedure planning methods, we additionally consider the proportion of instances that require intermediate visual observations as an indicator of cost. Where applicable, we keep the cost identical when comparing the accuracy metrics of different configurations for fairness.

**Baseline.** We adopt two baseline models, namely PDPP (Wang et al., 2023) and MDPP (Fang et al., 2023) to implement our active procedure planning approach. Both models perform passive planning, which gives the baseline performance. It should be noted that due to access to additional visual observations, our method enjoys more information than the baselines. As such, a direct comparison between our method and baselines is unfair. To overcome this issue, we implement enhanced baselines by re-training the passive learning models with additional visual observations as in our active planning approach. In particular, we randomly select a set of samples that are enhanced with additional visual observations, where the numbers of instances enhanced with visual observations are kept identical to the active planning counterparts.

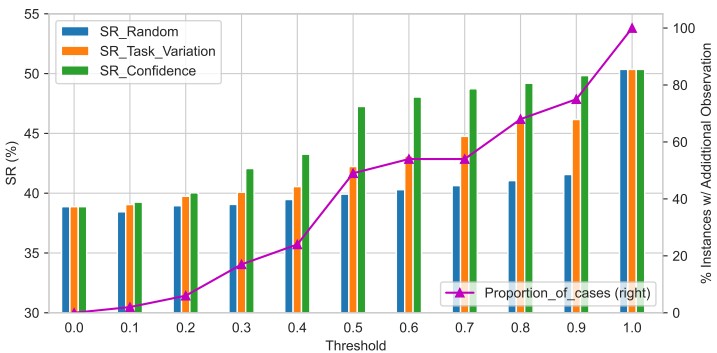

Figure 4: Success rate based on different uncertainty metrics (CrossTask, $T = 3$, MDPP model).

## 5.2 RESULTS

### 5.2.1 EFFECT OF UNCERTAINTY-AWARENESS IN DATA SAMPLING

We have proposed two uncertainty metrics, namely task sequence variation score (Eq. 1) and calibrated prediction confidence score (Eq. 2). An immediate question is which of them (if any) captures uncertainty better in terms of active sampling of visual observations. In addition, since the sampling output is controlled by pre-set thresholds, it is important to know what is the reasonable threshold.

First, we study the effect of uncertainty metrics (Eq. 4) using two configurations: (1) $w_1 = 1, w_2 = 0$, which models uncertainty based on task sequence variation, and (2) $w_1 = 0, w_2 = 1$, which models uncertainty based on calibrated confidence score. We use them to sample visual observations and perform procedure planning based on the masked diffusion for procedure planning (MDPP) model (Fang et al., 2023). To ensure an equivalent number of instances being augmented with additional visual observations in both configurations, we adopt different thresholds for each configuration. In particular, we first set the uncertainty threshold $\tau$ for configuration (1) so that $\tau$ is in the range of $[0, 1]$ with an interval of $0.1$. We test the performance of the models on these thresholds while keeping a record of the number of intermediate visual observations being included. Next, we set thresholds for configuration (2) to keep the proportions of instances being augmented with visual observations identical to the respective conditions in (1). Note that $\tau = 0$ corresponds to pure passive procedure planning in the baseline model, and $\tau = 1$ means additional observations are made for all instances. For benchmarking, we further adopt a random baseline where an equivalent number of visual observations is added based on random sampling.

Fig. 4 shows the comparison of success rate on the CrossTask dataset with planning horizon $T = 3$. Not surprisingly, when more visual observations are added (indicated by the curve in magenta), performance (*SR*) increases from about $38.9\%$ ($\tau = 0$) to $50.4\%(\tau = 1)$, irrespective of the sampling methods. However, when $0 < \tau < 1$, the proposed uncertainty-based sampling methods lead to higher *SR* increases compared to random sampling. In fact, the performance gain of random sampling is almost negligible until there are more than $70\%$ instances being augmented. In comparison, both uncertainty-based sampling methods lead to a consistent increase in *SR* even with a small number of augmented instances. Moreover, such performance gain is more evident in uncertainty measured from calibrated confidence score than from task variation score. In fact, starting from $\tau = 0.3$ (which translates to about $17\%$ instances being augmented), the boosting effect is substantially higher with the calibrated confidence score than with the task variation score. Similar results can be seen on *mAcc* and *mIoU* (refer to Appendix A.1).

Next, to further understand the relative importance of these two uncertainty metrics, we conduct sensitivity analysis with different combinations of relative weights. Fig. 5 shows the results of different weight combinations evaluated on CrossTask with $T = 3$ (refer to Appendix A.2 for a comprehensive ablation study). It is shown that using only task variation for uncertainty evaluation leads to the worst performance (orange line). On the other hand, using confidence score alone as the uncertainty metrics gives competitive outcomes although slightly inferior to other weight

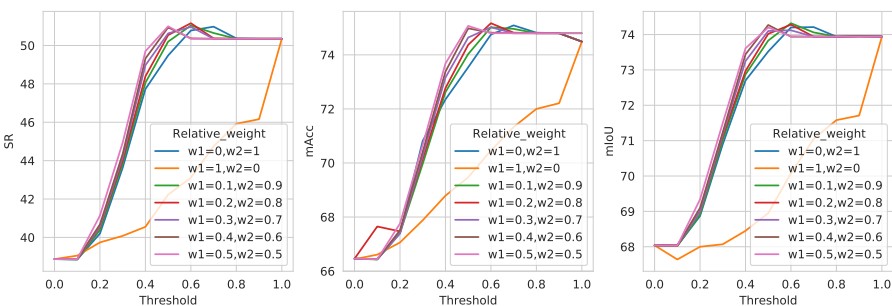

Figure 5: Performance evaluation based on different relative weights of uncertainty metrics.

assignments for $\tau < 0.6$. Since a smaller threshold is favorable (indicating fewer samples needed for observation, and hence lower cost), combining task variance and confidence score seems to be a better choice. The results also show that the outcome is not particularly sensitive to different relative weights as long as both metrics are included. Therefore, in subsequent experiments, we adopt three thresholds $\tau = \{0.3, 0.4, 0.5\}$, which generally maintain a good balance of performance gain and observation cost. Interestingly, the highest accuracy is typically achieved when $\tau = \{0.5, 0.6, 0.7\}$, which is counter-intuitive since one may expect that $\tau = 1$ (*i.e.*, all instances receive additional visual observation) engenders highest accuracy. The reason for this phenomenon is that in some instances, the start and goal state observations are informative enough to reason about the procedure plan. Adding an additional intermediate observation may not provide useful information. In fact, if such information is not consistent with that from initial observations, it may cause the model to mis-classify the procedure plan, which would otherwise be predicted correctly. With partial observation guided by uncertainty, our model refrains from full observation, thus reducing the possibility of disturbing correct predictions. This underscores the importance of judicious uncertainty estimation.

### 5.2.2 EVALUATION ON DIFFERENT PLANNING HORIZONS

We further study the effect of uncertainty-awareness in different planning horizons, *i.e.*, $T = \{3, 4, 5, 6\}$, evaluated on two base models, *i.e.*, PDPP and MDPP. We assign equal weights to two uncertainty metrics and adopt sampling thresholds $\tau = \{0, 0.3, 0.4, 0.5, 1\}$ based on the previous discussion. Again, we adopt the random sampling method to get baseline performance.

Table 1 shows the results with $T = \{3, 4\}$ on CrossTask. The boosting effect of active sampling is consistent in both planning horizons and across different evaluation metrics. For example, when $T = 3$, there is nearly 5% increase in *SR* with PDPP_0.3 and MDPP_0.3, compared to PDPP_0 and MDPP_0, respectively. Importantly, uncertainty-aware sampling is advantageous to random sampling by a large margin in all experiment conditions. Table 2 further shows results of success rate with $T = \{3, 4, 5, 6\}$ on the CrossTask dataset and $T = \{3, 4\}$ on the COIN dataset. The effect of uncertainty-based active sampling is more obvious on CrossTask, *i.e.*, it achieves larger performance gain compared to passive planning and random sampling baselines. Such an effect is marginal on the COIN dataset. This can be explained by the fact that there is less task variation (uncertainty) in the COIN dataset than in CrossTask (Zhao et al., 2022; Wang et al., 2023). Results on *mAcc* and *mIoU* for long-horizon planning are available in Appendix A.3 (CrossTask) and Appendix A.4 (COIN).

### 5.2.3 QUALITATIVE RESULTS

To illustrate the effect of active planning using uncertainty estimation, we show an example in Fig. 6. The task *Make Lemonade* has a planning horizon of $T = 5$. There are multiple variant plans for this task, and the one generated by the passive procedure planning model (middle row) is plausible but incorrect. Our uncertainty evaluation model predicts an uncertainty score of 0.708, which fairly reflects the potential of multiple plans in need of scrutiny. Hence, an additional observation is performed (frame in red box). With the additional observation, the model can predict the correct sequence of actions, while also resulting in a reduced uncertainty score of 0.385. The example also demonstrates a reasoning process that is more transparent and understandable by humans, which is important to foster trust in the model. More visualization examples are available in Appendix A.8.

Table 1: Performance of benchmarks with planning horizons T∈{3, 4} on CrossTask. PDPP_0.3 means that it adopts PDPP framework (Wang et al., 2023) with the uncertainty threshold set to 0.3. Threshold "0" means no additional observation is made and "1" means all instances have one additional observation. Numbers in brackets are the performance of random sampling.

| Models | T=3 | | | T=4 | | |
|---|---|---|---|---|---|---|
| | SR | mAcc | mIoU | SR | mAcc | mIoU |
| PDPP_0 | 37.2 | 64.7 | 66.6 | 21.5 | 57.8 | 65.1 |
| PDPP_0.3 | 42.0(39.0) | 67.8(66.1) | 69.8(68.0) | 25.0(22.8) | 61.6(60.3) | 66.5(55.8) |
| PDPP_0.4 | 45.1(40.0) | 70.3(66.8) | 71.5(68.5) | 28.9(24.1) | 64.4(61.6) | 68.3(66.6) |
| PDPP_0.5 | 48.2(40.4) | 72.3(67.4) | 72.9(68.7) | 31.0(25.4) | 66.0(62.3) | 70.3(67.3) |
| PDPP_1.0 | 49.83 | 74.0 | 73.4 | 32.9 | 62.3 | 71.6 |
| MDPP_0 | 38.9 | 66.5 | 68.0 | 23.5 | 60.2 | 66.8 |
| MDPP_0.3 | 43.6(39.7) | 70.8(67.2) | 70.9(68.5) | 26.5(23.8) | 63.3(61.2) | 68.9(67.0) |
| MDPP_0.4 | 47.7(41.0) | 72.4(68.2) | 72.7(69.2) | 30.1(24.5) | 65.8(62.0) | 70.5(67.3) |
| MDPP_0.5 | 49.5(41.4) | 73.5(68.7) | 73.5(69.5) | 32.3(26.0) | 67.5(62.9) | 71.4(68.0) |
| MDPP_1.0 | 50.4 | 74.5 | 74.0 | 34.1 | 69.3 | 72.0 |

Table 2: Performance (SR) comparison for long-horizon planning on two datasets.

| | CrossTask | | | | COIN | |
|---|---|---|---|---|---|---|
| | T=3 | T=4 | T=5 | T=6 | T=3 | T=4 |
| PDPP_0 | 37.2 | 21.5 | 13.6 | 8.5 | 21.3 | 14.4 |
| PDPP_0.3 | 42.0(39.0) | 25.0(22.8) | 15.7(14.1) | 10.9(9.0) | 21.6(21.0) | 14.5(14.4) |
| PDPP_0.4 | 45.1(40.0) | 28.9(24.1) | 17.6(14.2) | 12.4(9.5) | 22.0(21.1) | 14.7(14.6) |
| PDPP_0.5 | 48.2(40.4) | 31.0(25.4) | 18.7(14.9) | 13.7(10.0) | 22.0(21.4) | 14.7(14.7) |
| PDPP_1.0 | 49.83 | 32.9 | 19.8 | 14.3 | 21.9 | 14.8 |
| MDPP_0 | 38.9 | 23.5 | 15.3 | 10.1 | 29.4 | 21.4 |
| MDPP_0.3 | 43.6(39.7) | 26.5(23.8) | 16.5(14.4) | 11.6(9.9) | 31.3(30.3) | 22.7(22.2) |
| MDPP_0.4 | 47.7(41.0) | 30.1(24.5) | 19.8(15.0) | 13.5(10.2) | 31.8(30.8) | 23.0(22.5) |
| MDPP_0.5 | 49.5(41.4) | 32.3(26.0) | 22.1(16.4) | 15.3(11) | 32.0(31.3) | 23.0(22.8) |
| MDPP_1.0 | 50.4 | 34.1 | 24.1 | 15.8 | 32.1 | 23.1 |

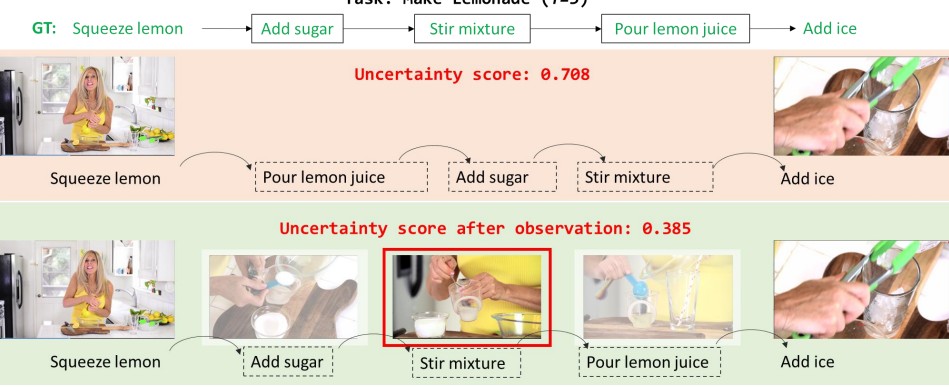

Figure 6: Visualization of uncertainty-based visual observation to resolve ambiguity.

## 6 CONCLUSIONS

We formulate an active procedure planning problem by enabling selective visual observations during the inference stage of procedure planning in instructional videos. The purpose is to resolve ambiguities due to inadequate visual representations and the probabilistic nature of the problem. We propose an uncertainty-aware active planning methodology based on well-crafted uncertainty metrics derived from the model's prediction confidence and task variation characteristics. Experimental results show that the proposed method can substantially improve the accuracy of generated plans while keeping the observation cost low. The approach is a step towards building trusted and explainable AI that allows agents to actively explore the world and assist humans. Although the interactivity in the current problem settings, *i.e.*, instructional videos, is quite limited, the concept of uncertainty-based active planning and the empirical evidence of its effectiveness are useful for developing solutions for more complicated problems.

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

## A    ACTIVE PROCEDURE PLANNING WITH UNCERTAIN-AWARENESS IN INSTRUCTIONAL VIDEOS: APPENDIX

### A.1    EFFECT OF UNCERTAINTY METRICS IN TERMS OF MACC AND MIOU

We have shown the effect of different uncertainty metrics, namely task sequence variation and prediction confidence, on the success rate (*SR*) in the main manuscript. We use the same strategy to evaluate such effect on the other two evaluation metrics, namely *mAcc* and *mIou*. Fig. 7 and Fig. 8 show the performance outcome in relation to uncertainty thresholds and the resulting proportion of samples with additional observations. We observe similar trends to the *SR* performance. (1) Both uncertainty-based methods outperform random sampling baseline. (2) Performance increases with more observations made. (3) The calibrated confidence score is more effective in capturing uncertainty based on the greater magnitude of performance improvement. The results echo the findings in the main manuscript.

### A.2    ABLATION STUDY ON UNCERTAINTY METRICS WITH DIFFERENT PLANNING HORIZONS

In Section 5.2.1, we show the relative importance of two uncertainty metrics with planning horizon $T = 3$. For a more comprehensive ablation study, we evaluate such an effect for longer planning horizons $T = \{4, 5, 6\}$. We adopt three relative weight assignments, namely (1) $w_1 = 1, w_2 = 0$, (2) $w_1 = 0, w_2 = 1$, and (3) $w_1 = w_2 = 0.5$. Similar to before, (1) and (2) represent uncertainty evaluated as only task variation score and calibrated prediction confidence score, respectively; (3) denotes a more balanced weight assignment. We choose thresholds $\tau = \{0.3, 0.4, .05, .06\}$ based on previous results to avoid tedious computing. Fig. 9 shows the outcome.

The result is consistent with findings reported in the main manuscript. First, the calibrated confidence score seems to capture uncertainty better than the task variation score in terms of boosting prediction accuracy. Second, combining both uncertainty metrics generally facilitate better outcome than using one metric alone, indicating the positive contribution of both metrics. However, the effect of the task variation score diminishes as the planning horizon goes longer, *e.g.*, at $T = 6$, where the calibrated confidence score alone can give performance that is equivalent to or even better than combined metrics under some threshold levels.

### A.3    PERFORMANCE OF LONG-HORIZON PLANNING ON THE CROSSTASK DATASET

In Table 3, we show more results for long-horizon planning $T = \{5, 6\}$ on the CrossTask dataset, as an extension to Table 1. As expected, the accuracy reduces as the planning horizon becomes longer. However, the boosting effect of additional observation is consistent across different evaluation metrics, for different baseline frameworks (*i.e.*, PDPP and MDPP) and with different threshold settings.

### A.4    EVALUATION ON THE COIN DATASET

We show the performance on the COIN dataset with planning horizon $T = \{3, 4\}$, following conventions in (Zhao et al., 2022; Wang et al., 2023). In Table 4, one can see that there is a slight performance gain due to additional visual observations guided by the uncertainty metrics. However, the magnitude of gain is marginal and disparate for the two frameworks. In fact, for the PDPP framework (Wang et al., 2023), additional observations do not lead to a consistent increase in accuracy when samples are randomly selected, as shown by the figures in the random baseline. For our active planning method, the increase is quite small, *e.g.*, when $T = 3$, there is about $0.7\%$ increase in *SR* with PDPP_0.5 compared to PDPP_0. The boosting effect is more evident for the MDPP framework, reaching $2.6\%$ increase in *SR* for MDPP_0.5 compared to MDPP_0. We suspect that the COIN dataset involves fewer task variations and uncertainty compared to CrossTask as reported in previous works (Bi et al., 2021; Zhao et al., 2022; Wang et al., 2023). Therefore, the effect of additional observation is less significant on the COIN dataset.

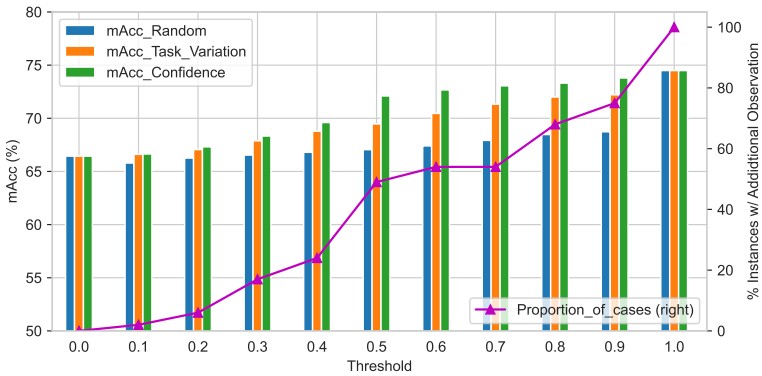

Figure 7: Mean accuracy (*mAcc*) based on different uncertainty metrics (CrossTask, $T = 3$, MDPP model).

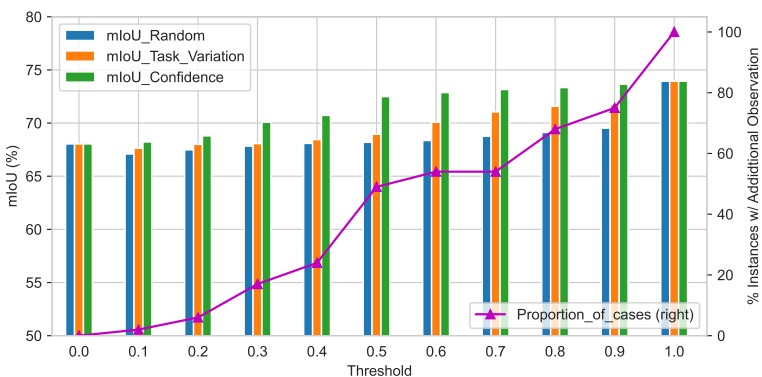

Figure 8: Mean intersection over union (*mIoU*) based on different uncertainty metrics (CrossTask, $T = 3$, MDPP model).

Table 3: Performance of benchmarks with planning horizons $T \in \{5, 6\}$ on CrossTask. PDPP_0.3 means to adopt the PDPP model (Wang et al., 2023) with an uncertainty threshold of 0.3. Threshold "0" means that no additional observation is used, and "1" means that all instances have one additional observation. Numbers in brackets indicate the performance of random sampling.

| Models | T=5 | | | T=6 | | |
|---|---|---|---|---|---|---|
| | *SR* | *mAcc* | *mIoU* | *SR* | *mAcc* | *mIoU* |
| PDPP_0 | 13.6 | 54.1 | 65.3 | 8.5 | 50.1 | 65.4 |
| PDPP_0.3 | 15.7(14.1) | 56.7(55.0) | 65.9(65.4) | 10.86(9.0) | 54.8 (51.4) | 67.6(66.0) |
| PDPP_0.4 | 17.6(14.2) | 59.0(55.3) | 66.6(65.3) | 12.4(9.5) | 57.3(52.4) | 68.7(66.3) |
| PDPP_0.5 | 18.7(14.9) | 60.3(55.9) | 67.0(65.2) | 13.7(10.0) | 59.0(53.0) | 69.5(66.5) |
| PDPP_1.0 | 19.9 | 61.9 | 67.2 | 14.3 | 60.5 | 69.8 |
| MDPP_0 | 15.3 | 56.2 | 66.0 | 10.1 | 52.0 | 66.1 |
| MDPP_0.3 | 16.7(14.4) | 58.9(56.4) | 67.8(65.9) | 11.6(9.9) | 55.6(52.6) | 57.5(65.9) |
| MDPP_0.4 | 19.8(15.0) | 61.6(57.2) | 69.1(66.4) | 13.5(10.2) | 58.1(53.4) | 68.7(66.3) |
| MDPP_0.5 | 22.2(16.4) | 63.5(58.1) | 70.2(66.8) | 15.3(11.0) | 60.2(54.5) | 69.9(66.6) |
| MDPP_1.0 | 24.1 | 65.6 | 70.2 | 15.8 | 61.5 | 70.1 |

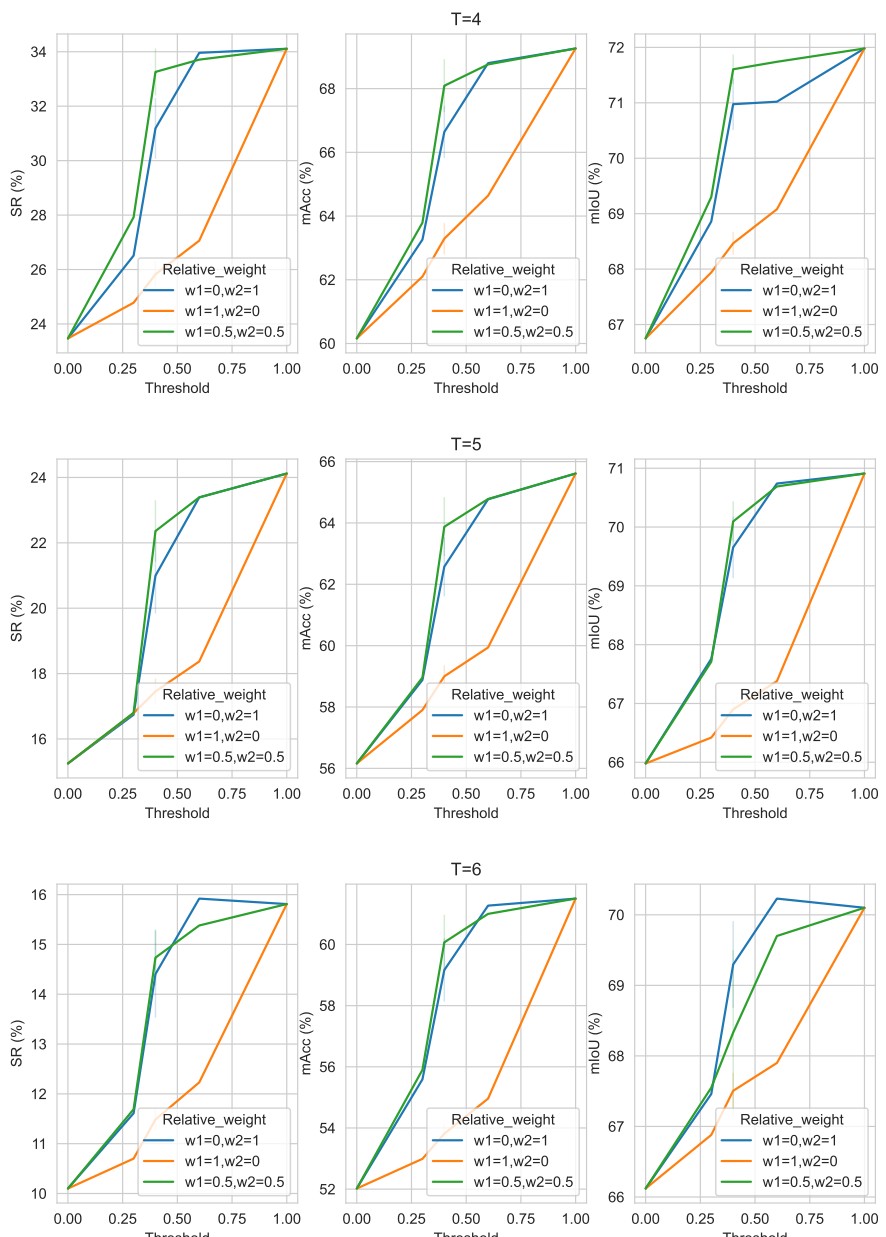

Figure 9: Ablation study on uncertainty metrics for long-horizon planning (CrossTask, $T = 4, 5, 6$, MDPP model).

## A.5 HYPER-PARAMETER SETTING FOR EACH DATASET

During the training stage, we implement a linear warm-up scheme for adjusting learning rate settings to align with the specific characteristics of various datasets. The CrossTask dataset starts with a linear warm-up for 4,000 steps until it reaches 5e-4. Following this warm-up phase, the learning rate undergoes decay by a factor of 0.5 at the $10,000^{th}$, $16,000^{th}$, and $22,000^{th}$ steps. For the COIN dataset, given its substantial scale, we progressively ramp up the learning rate to 1e-5 over the course of 4,000 training steps. Then, 0.5 decay is applied at the $14,000^{th}$ and $24,000^{th}$ steps. Following this initial increase, the learning rate remains constant at 2.5e-6 for the remainder of the training

Table 4: Performance of benchmarks with planning horizons $T \in \{3, 4\}$ on COIN. PDPP_0.3 means to adopt the PDPP model (Wang et al., 2023) with an uncertainty threshold of 0.3. Threshold "0" means that no additional observation is used, and "1" means that all instances have one additional observation. Numbers in brackets indicate the performance of random sampling.

| Models | T=3 | | | T=4 | | |
|---|---|---|---|---|---|---|
| | SR | mAcc | mIoU | SR | mAcc | mIoU |
| PDPP_0 | 21.3 | 45.6 | 51.8 | 14.4 | 44.1 | 51.4 |
| PDPP_0.3 | 21.6(21.0) | 46.6(46.2) | 51.8 (51.5) | 14.5(14.4) | 44.6(44.4) | 51.6(51.6) |
| PDPP_0.4 | 22.0(21.1) | 47.0(46.4) | 51.9(51.5) | 14.7(14.6) | 44.6(44.6) | 51.9(51.8) |
| PDPP_0.5 | 22.0(21.4) | 47.1(46.7) | 51.9(51.7) | 14.7(14.7) | 44.6(44.7) | 51.9(51.7) |
| PDPP_1.0 | 21.9 | 47.1 | 51.9 | 14.8 | 44.7 | 51.9 |
| MDPP_0 | 29.4 | 49.5 | 52.2 | 21.3 | 46.8 | 52.5 |
| MDPP_0.3 | 31.3(30.3) | 53.1(52.4) | 59.6(59.3) | 22.7(22.2) | 50.0(49.8) | 59.7(59.6) |
| MDPP_0.4 | 31.8(30.8) | 53.4(52.8) | 59.7(59.4) | 23.0(22.5) | 50.4(50.1) | 59.9(59.7) |
| MDPP_0.5 | 32.0(31.3) | 53.7(53.2) | 59.7(59.5) | 23.0(22.8) | 50.4(50.3) | 59.8(59.8) |
| MDPP_1.0 | 32.1 | 53.8 | 59.7 | 23.1 | 50.5 | 59.9 |

iterations. The batch size is set as 256 for both datasets. The training epochs on CrossTask and COIN datasets are 120 and 800, respectively.

## A.6 UNCERTAINTY CHARACTERISTICS OF THE TASK SEQUENCE VARIATION

We show the distribution of the task variation score used as the uncertainty metric on two datasets with different planning horizons (Fig. 10). On both datasets, there is a significant amount of instances with near "0" scores, indicating the existence of a dominant trajectory and minimal variation for the respective tasks. The proportion of instances with near "0" score is even higher on the COIN dataset than on CrossTask, which is consistent with previous evidence that CrossTask involves higher uncertainty in the procedure plans (Zhao et al., 2022; Wang et al., 2023). Moreover, the task variation score generally becomes higher with a longer planning horizon as seen from the relatively higher proportion of instances with larger task variation scores in $T = 5, 6$ relative to $T = 3, 4$ on CrossTask; and similarly for $T = 4$ *vs.* $T = 3$ on COIN.

## A.7 EFFECT OF TEMPERATURE SCALING FOR UNCERTAINTY CALIBRATION

We have adopted temperature scaling (Guo et al., 2017) as a mean to calibrate uncertainty based on the predicted confidence score. In this section, we show how this calibration affects procedure planning outcomes. Fig. 11 shows the evaluation metrics (*SR, mAcc, mIoU*) at different uncertainty threshold levels $[0, 1]$ with an interval of 0.1. The two curves show the change in the proportion of instances that receive an additional visual observation. We observe that (1) the trend of boosting effect owing to active sampling is identical with or without calibration; (2) the performance gain is slightly higher with uncertainty calibration using temperature scaling. This is largely attributed to the inclusion of more instances with additional visual observation for a specific threshold level. In that sense, temperature scaling does not directly affect performance. In fact, uncertainty calibration using temperature scaling does not change the classification output (*i.e.*, the class label). Rather it only changes the confidence level of the predictions. As such, it modifies the amount of samples selected at the same threshold levels. We do see a more balanced and progressive increase in the number of samples selected as the threshold increases at the fixed internal 0.1, which is beneficial to the robustness of outcomes.

## A.8 VISUALIZATION OF PROCEDURE PLANS ENHANCED BY VISUAL OBSERVATIONS

We show examples of how uncertainty-based visual observations affect the procedure planning for different planning horizons, *i.e.*, 3, 4, 5, and 6. In all the examples (Fig. 12 - Fig. 15), we show the ground-truth procedure plan in green. Procedure plans generated without intermediate visual observations are in the boxes shaded in pink color; and those with intermediate visual observations (images with red borders) are shaded in lime green. The uncertainty threshold is set at 0.5 in all cases. We show the initial uncertainty scores and updated ones after additional visual observations.

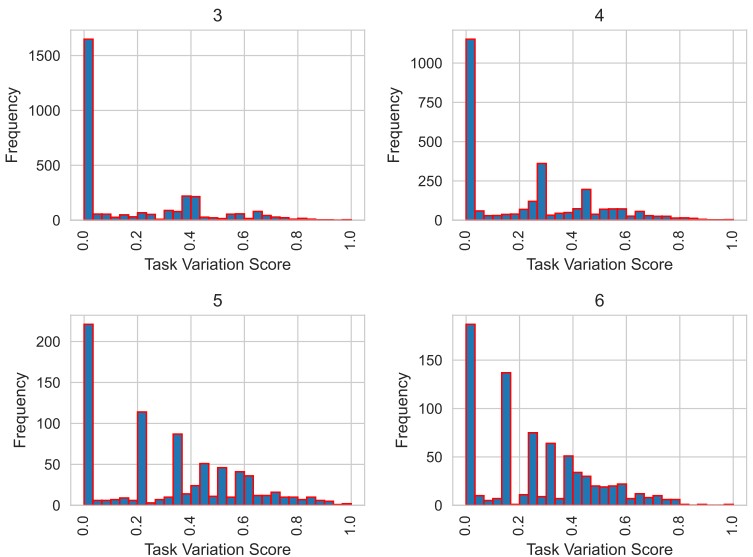

(a) Histogram of uncertainty based on the task variation score at different planning horizons ($T = 3, 4, 5, 6$) on the CrossTask dataset

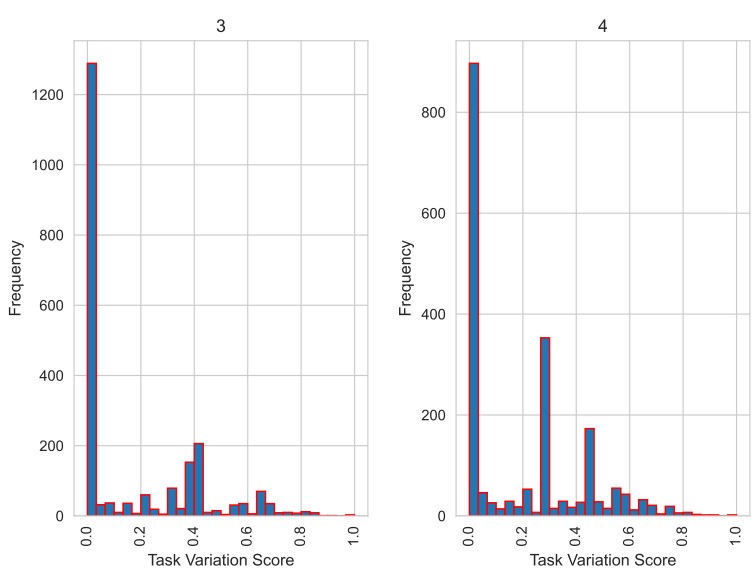

(b) Histogram of uncertainty based on the task variation score at different planning horizons ($T = 3, 4$) on the COIN dataset

Figure 10: Distribution of the task variation score when using it as the uncertainty metric.

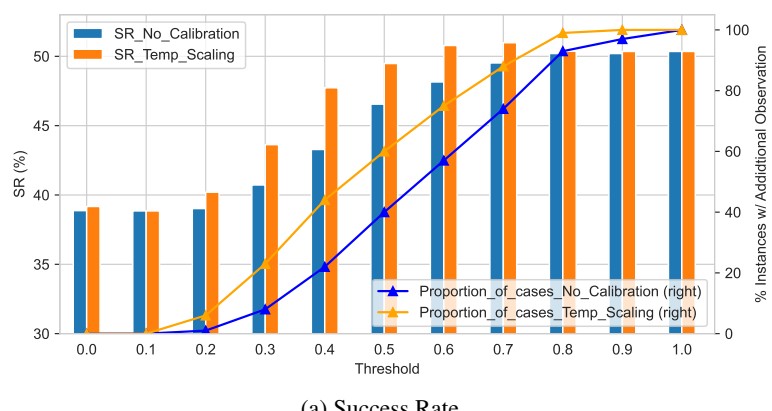

(a) Success Rate

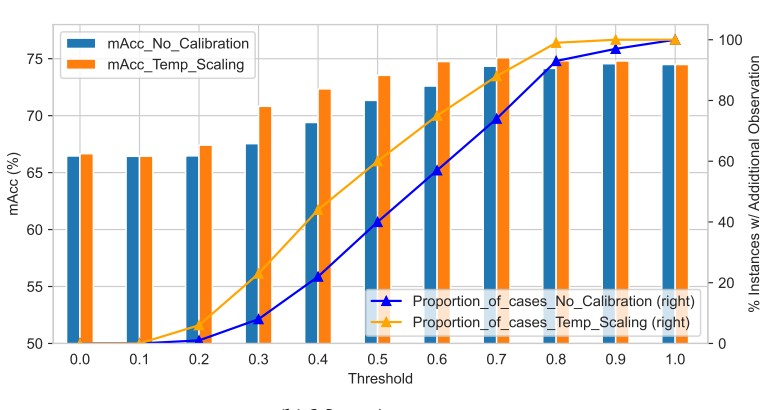

(b) Mean Accuracy

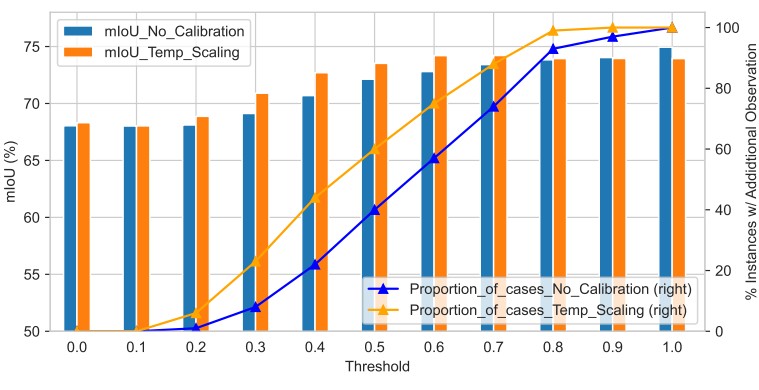

(c) Mean Intersection over Union

Figure 11: Effect of uncertainty calibration with temperature scaling (evaluated with the MDPP model).

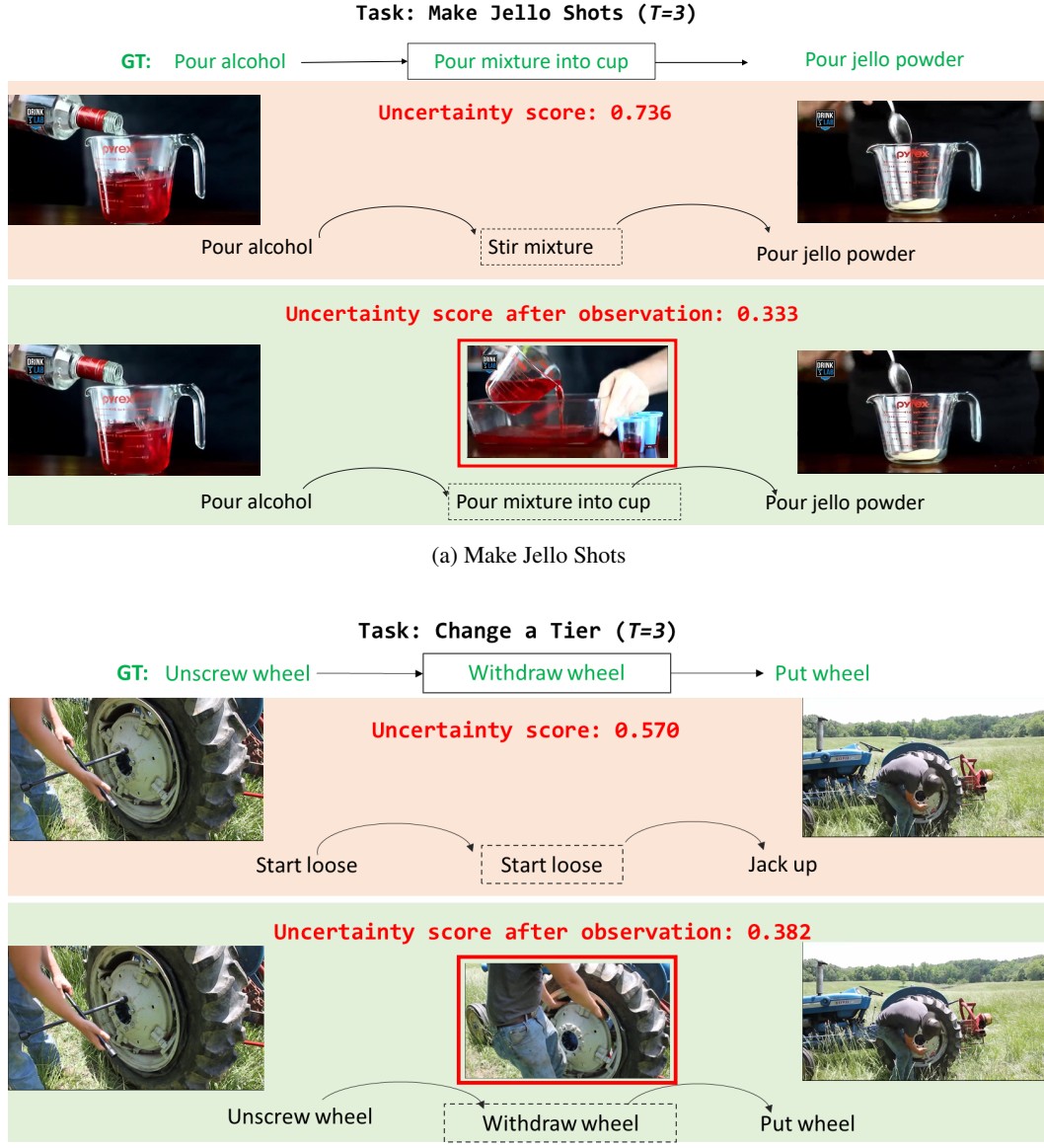

Figure 12: Visualization of using an uncertainty-based visual observation to resolve the ambiguity $(T = 3)$.

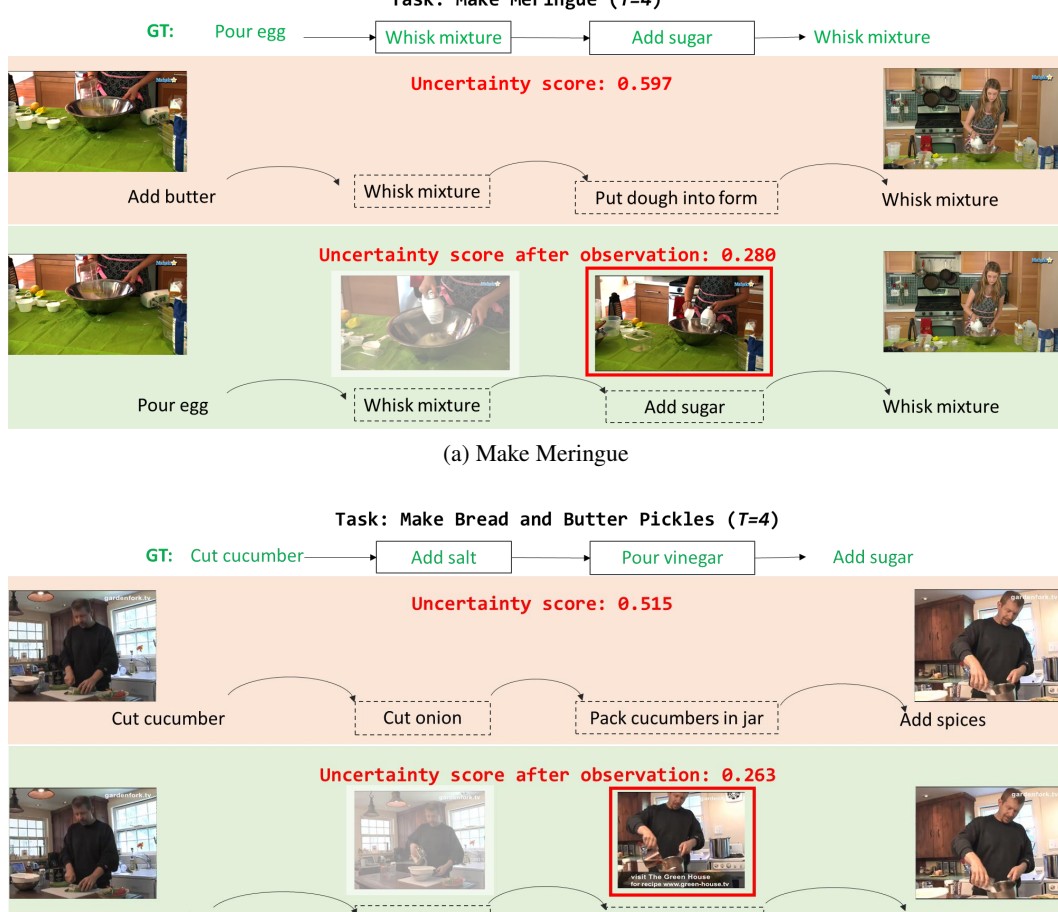

(a) Make Meringue

(b) Make Bread and Butter Pickles

Figure 13: Visualization of using an uncertainty-based visual observation to resolve the ambiguity ($T = 4$).

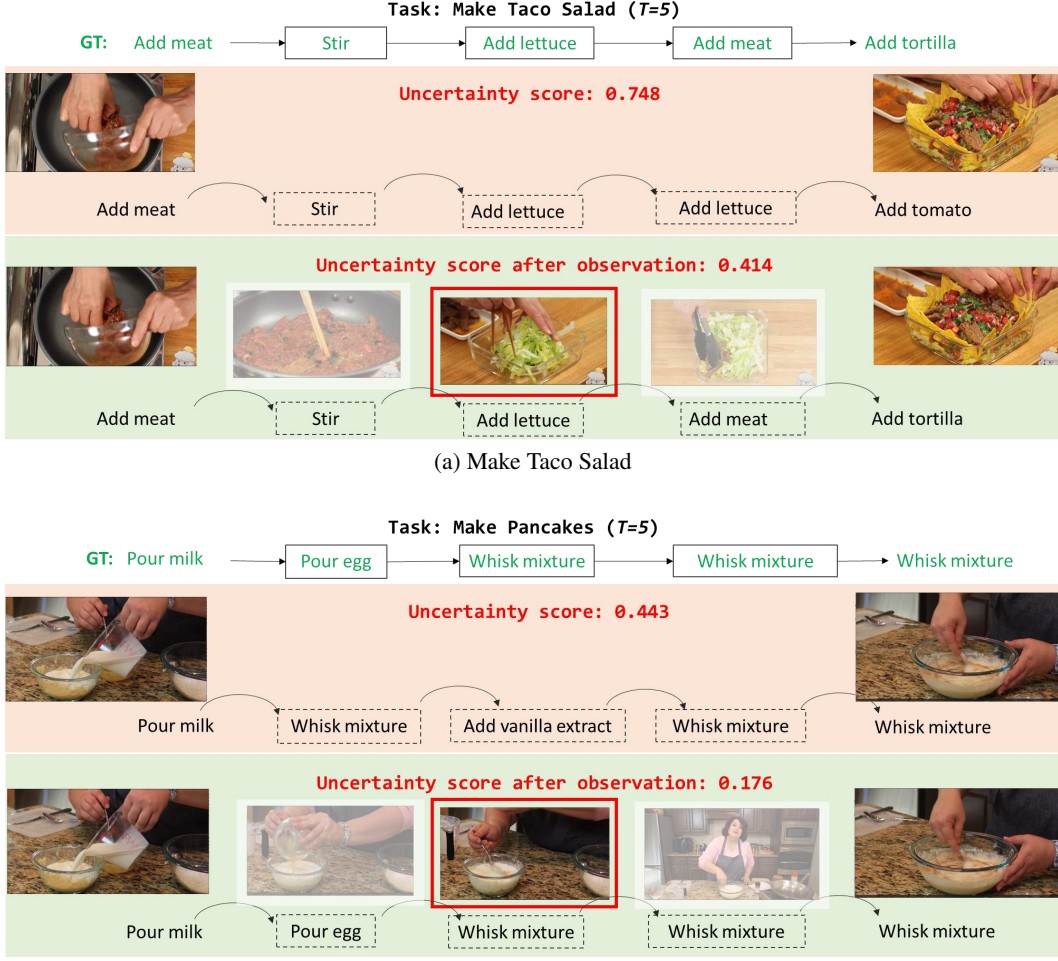

(a) Make Taco Salad

(b) Make Pancakes

Figure 14: Visualization of using an uncertainty-based visual observation to resolve the ambiguity ($T = 5$).

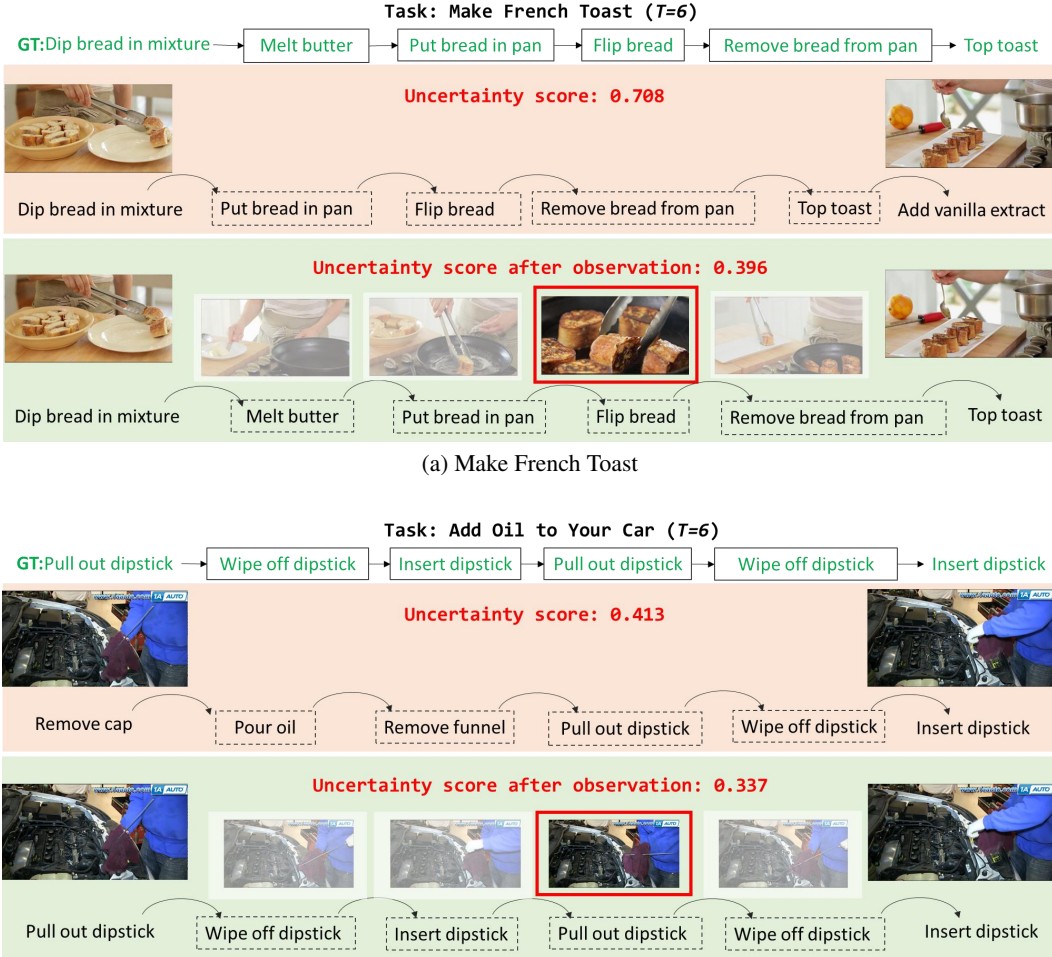

(a) Make French Toast

(b) Add Oil to Your Car

Figure 15: Visualization of using an uncertainty-based visual observation to resolve the ambiguity ($T = 6$).

