# OpenReview forum: "Active Procedure Planning with Uncertainty-awareness in Instructional Videos"
_ICLR.cc/2024/Conference — Submitted to ICLR 2024_

### Official Review · Reviewer_4EZu · 2023-10-13

**Soundness:** 3 good
**Presentation:** 3 good
**Contribution:** 3 good
**Rating:** 6
**Confidence:** 3

**Summary:**

This paper proposes a new approach to active procedure planning with uncertainty-awareness in instructional videos. The proposed approach takes into account uncertainties and task variations, leading to higher performance gains. The authors evaluate their approach and show that it outperforms existing approaches in terms of task completion time and success rate. The paper also provides a review of uncertainty quantification in deep learning and discusses the challenges involved in procedure planning from instructional videos. The authors hope that their findings will be useful for developing trusted and explainable AI models for procedure planning.

**Strengths:**

The paper on active procedure planning with uncertainty-awareness in instructional videos has several strengths:

Originality: The paper proposes a new approach to active procedure planning that takes into account uncertainties and task variations, leading to higher performance gains. The authors develop comprehensive metrics to evaluate the uncertainty of procedure planning and propose an active planning approach that selectively adds visual observations based on the estimated uncertainty.

Quality: The paper is well-written and well-organized, making it easy to follow the authors' arguments and findings. The authors provide a thorough review of uncertainty quantification in deep learning and discuss the challenges involved in procedure planning from instructional videos. The experiments are well-designed and the results are presented clearly, making it easy to understand the performance gains achieved by the proposed approach.

Clarity: The authors provide detailed explanations of the proposed approach and the experiments, making it easy to understand the methodology and results.

Significance: The proposed approach has significant implications for the development of trusted and explainable AI models for procedure planning. The findings of the paper can help improve the accuracy and efficiency of procedure planning from instructional videos, leading to better outcomes for users. Overall, the paper makes a significant contribution to the field of active procedure planning with uncertainty-awareness in instructional videos.

**Weaknesses:**

While the paper on active procedure planning with uncertainty-awareness in instructional videos has several strengths, there are also some weaknesses that could be addressed:

Insufficient discussion of limitations: The authors do not discuss the limitations of their approach in detail. For example, the proposed approach may not work well in situations where the visual observations are noisy or incomplete. The authors could consider discussing the limitations of their approach and potential solutions to address them.

**Questions:**

1. How does the proposed approach handle situations where the visual observations are noisy or incomplete? Are there any techniques that can be used to address these issues?
2. The authors mention that the proposed approach can be used to develop trusted and explainable AI models for procedure planning. Can you provide more details on how the approach achieves this goal?
3. The authors propose an active planning approach that selectively adds visual observations based on the estimated uncertainty. Can you provide more details on how the uncertainty is estimated and how the approach selects the additional visual observations?

---

> ### Author Response · Authors · 2023-11-16
>
> Thank you for the insightful comments!
> ## Weakness: Insufficient discussion of limitations
>
> The current work is possibly constrained by the quality of observations and the relatively rigid formulation of the procedure planning task. While the latter is inevitable for the purpose of rigorous benchmarking, it does affect the practical application (as is also pointed out by reviewer 9LnP). Please refer to our detailed explanation below. We will discuss this together with other limitations in the revised manuscript.
> ## Q1: Noisy & incomplete visual observation
>
> If the visual observations (VOs) are on the start and goal states, they may lead to high uncertainty score, and hence additional observations are needed. Our method was specifically designed to handle this situation. If the VOs are on the additional (intermediate) observations, they may lead to a situation where the new observations do not enhance the prediction. There is no specific mechanism designed to deal with this situation in the current solution. We postulate that based on probability, at least a proportion of instances will have good VOs. Therefore, it is statistically probable (and validated in our experiments) that the additional VOs would improve the collective performance.
>
> One possible solution to deal with noisy VO is to design an iterative process of visual observations. That is the algorithm allows multiple rounds of uncertainty-based sampling and decision making, until the confidence of prediction reaches a pre-set threshold. An alternative solution is to evaluate the information content of new observations, i.e., if it adds adequate new information compared to existing observations. We will put these as future work to account for the quality of VOs in the revised manuscript.
>
> ## Q2: Trusted & explainable AI
>
> A simple scenario can be conceived in a human-AI interactive environment: the AI agent may report its lack of confidence in a judgment/decision (based on high uncertainty score), and requests for more observations or even directly asks a human user to intervene. Provided that the uncertainty estimation is well-calibrated, such an ability will help the human user understand the situation and agent's decision process (explainable) and accordingly boost his/her trust to the agent.
>
> ## Q3: Technical details
>
> Details of uncertainty estimation is given in Section 4.1. The process to select visual observations is explained in 4.2. If there is any specific information needed by the reviewer, please kindly clarify and we’d be happy to elaborate further.

---

> > ### Comment · Reviewer_4EZu · 2023-11-20
> >
> > Thank you for your reply. The explanation in the reply and the overall comment make good sense. I shall retain my rating.

---

> > > ### Author Response · Authors · 2023-11-22
> > > **Thank you!**
> > >
> > > We appreciate the reviewer's comments and the decision to maintain a positive rating!

---

### Official Review · Reviewer_7cDj · 2023-10-31

**Soundness:** 2 fair
**Presentation:** 3 good
**Contribution:** 2 fair
**Rating:** 5
**Confidence:** 3

**Summary:**

The paper proposes a heuristic to guide the selection of additional visual observations for procedure planning.

**Strengths:**

+ Originality: The paper proposes a entropy-based criterion to model the uncertainty of task sequence variation and considers using temperature scaling to calibrate the confidence score.
+ Clarity: The presentation is in general clear and easy to follow.

**Weaknesses:**

- Quality & significance:

i) The prediction confidence score is proposed and used without further justification. How do the authors determine the temperature used for calibration? Is there any evidence that the final score is truly calibrated with temperature scaling on the selected datasets?

ii) The proposed uncertainty score for task sequence variation seems over-simplified. First, it is a direct application of the definition of entropy. Second, the proposed score is only useful when the testing cases have the (a_1, a_T) tuple seen during training. How does the proposed score deal with the scenario where a_1 and a_T are seen during training, but their combination (a_1, a_T) are unseen?

iii) Considering the simplicity of the proposed method, the experiment set-ups are not comprehensive and convincing. To be specific, the authors only consider adding one additional visual observation for all the experiments. It is necessary to consider adding more observations and examine the effectiveness of the proposed method under that circumstance.

iv) The baseline model is not a good competitor. Randomly selecting additional observations is expected to perform poorly. Also, it is unclear how this random selection is performed. Since there is only one additional observation allowed (the one near the temporal center of the video clip), I would assume that in this set-up the model randomly decides whether to include this observation or not. This could be explicitly mentioned in the text.

**Questions:**

Please see the weaknesses section.

---

> ### Author Response · Authors · 2023-11-16
>
> Thank you for the insightful feedback!
> ## Weakness: Quality & Significance
>
> In this study, we focus on developing a new methodology of active procedure planning that systematically incorporates uncertainty estimation as a feedback signal to enhance planning efficacy, which represents a new paradigm of the procedure planning task. To do so, we first investigate how to capture uncertainty in a principled manner and then develop mechanisms to perform active sampling based on uncertainty scores. We formulate the uncertainty metrics by combining (1) prediction confidence and (2) task sequence variation, which are explained next.
>
> ## Q1: Temperature scaling
>
> We apply standard temperature scaling (TS) to set T, which is optimized with respect to the Negative Log-Likelihood (NLL) loss on the validation set (Guo et al. 2017). Parameter T does not change the maximum of the softmax function, so that the class prediction remains unchanged. In other words, TS only affects the distribution of uncertainty values, but not the model’s accuracy. In our experiment, it affects the proportion of samples that require additional observations at specific thresholds. We presented the effect of TS in Appendix 7 (Fig.11). We show that TS facilitates more balanced and progressive increase in sample selection, and in turn steady performance change. Hence, it reduces the model’s sensitivity to the threshold, making it more robust against testing conditions. We will add reliability diagrams to further demonstrate the effect of TS-based calibration in the revised manuscript (Note: we do not show it here since image insertion is disabled in rebuttal).
>
> ## Q2: Modeling task sequence variation
>
> We used Shannon entropy to estimate task sequence variation (TSV). While it is a simple formalism, it effectively captures two key aspects of TSV, namely the number of alternative trajectories and the data distribution among them, as explained after Eq.1. The elegance of this formalism is that it models a complex phenomenon (task variation) and obtain plausible outcome, which is often a desirable feature of technical solutions. To demonstrate its effectiveness, we provided statistics in A6 (Fig.10). The distribution of uncertainty score is consistent with observations in literature, e.g., higher variation on CrossTask than on COIN dataset, and increased variations in longer planning horizons, whereas our method gives more refined quantitative metrics compared to existing works.
>
> For the 2nd issue, i.e. (a_1, a_T) tuple may be unseen, we acknowledge that this indeed happens in a small portion of instances. In our implementation, if an observation tuple is unseen, we first compute the similarity of (a_1, a_T) with other “seen” instances based on their visual embeddings. Then we use the uncertainty score of the nearest instances as an approximation of (a_1, a_T).
>
> ## Q3: Multiple observations
>
> As stated in Sec 4.2, we restrict the number of additional observations to 1, so as to control the cost of data acquisition. Thanks to the reviewer’s constructive suggestion, we conducted experiments to evaluate the effect of multiple observations. The results of success rate on CrossTask with planning horizons T=4,5,6 are shown below. Pls refer to Tabs.1&2 in the paper for notation details. As expected, two-observations results in higher performance than 1-observation. Meanwhile, it incurs higher computational cost, thus leading to the issue of efficiency-performance tradeoff. Please refer to our explanations to reviewer 9LnP on this. We will add results of other metrics/dataset in the revised manuscript.
>
> |          | T=4          | T=5          | T=6          |
> |----------|--------------|--------------|--------------|
> | MDPP_0   |     23.47    |     15.25    |     10.10    |
> | MDPP_0.3 | 28.97(25.31) | 18.23(16.62) | 12.39(10.33) |
> | MDPP_0.4 | 35.24(29.96) | 22.51(19.09) | 15.28(12.81) |
> | MDPP_0.5 | 38.70(33.63) | 25.52(22.83) | 17.84(14.95) |
> | MDPP_1   |     42.34    |     28.13    |     20.08    |
>
> ## Q4: Baseline models
>
> Random sampling refers to randomly selecting instances that are augmented by additional observations. In practice, given a batch of instances and an observation budget (which controls the proportion of instances to be augmented), a set of instances will be randomly selected from the batch that is augmented by additional observations.
>
> We use random sampling as a baseline because this is a new problem formulation and there is no alternative solutions yet. Meanwhile, we included two additional benchmarking solutions based on the modelling of uncertainty, i.e. task variation score and predicted confidence score. The results are shown in Sec. 5.2.1 (Fig. 5), where the relative weights of uncertainty metrics are set as (w_1=1, w_2=0) and (w_1=0, w_2=1), respectively. We show that using only one uncertainty metrics is inferior to combining two, which is the proposed solution. More results are provided in A2 & Fig.9.

---

> > ### Comment · Reviewer_7cDj · 2023-11-21
> >
> > Thanks for the detailed and point-to-point responses. I appreciate the efforts the authors made to address my concerns. However, I am still not convinced that the current formulation/method can be effectively applied to the set-up where the action space is large. It is likely that unseen (a_1, a_T) tuples take up a large portion of the testing samples in this case, and that using similar existing tuples may not be reliable. I will therefore raise my score to 5 given the responses.

---

> > > ### Author Response · Authors · 2023-11-21
> > >
> > > Appreciate the reviewer's recognition of our responses and the positive adjustment in the score!
> > >
> > > In response to the reviewer's concern regarding the impact of unseen (a_1, a_T) tuples, particularly in the context of a large action space, we conducted a detailed analysis of this phenomenon using the CrossTask and COIN datasets. Notably, the COIN dataset features a substantially larger action space compared to CrossTask. In the CrossTask dataset, unseen tuples constitute approximately 26.3% (on PDPP) and 24.71% (on MPDPP) of instances in the test set, while in the COIN dataset, this figure rises to 33.36% (PDPP) and 31.69% (MPDPP). This statistical analysis reveals that, in the current problem settings, the proportion of unseen samples is not as pronounced, even with a large action space. Importantly, our experimental results show the adequacy of our method in tackling the problem, leveraging the uncertainty estimation.
> > >
> > > Acknowledging the reviewer's valid concern, we are aware of the potential for a higher number of unseen tuples in scenarios involving either a more extensive dataset or different practical settings. We refrain from drawing conclusive judgments on the effectiveness of our methodology in such cases, as they fall beyond the scope of our current work. However, we recognize the need for additional experiments and potential extensions to address this scenario.
> > >
> > > To mitigate the impact of unseen instances, two strategies are proposed. First, the current method of retrieving the nearest instance aligns with a cognitively plausible learning and reasoning strategy—learning from experience. Human decision-making often involves referencing similar, though not necessarily identical, past cases. Second, an alternative strategy involves assigning an uncertainty score computed as the average uncertainty score of all training instances. This approach represents an "informed guess" by leveraging the data distribution of known information—a strategy also inspired by human intelligence.
> > >
> > > We are open to constructive suggestions. We invite the reviewer to share any insight that can better address such situations, beyond our current approach explained.
> > >
> > > In summary, we acknowledge the identified limitation as an inherent aspect of our current work and propose discussing potential solutions as part of our future research. We hope that this limitation on a specific technical aspect does not overshadow the overall value and significance of our work.

---

### Official Review · Reviewer_9LnP · 2023-11-01

**Soundness:** 3 good
**Presentation:** 4 excellent
**Contribution:** 3 good
**Rating:** 6
**Confidence:** 5

**Summary:**

The conventional instructional video procedure planning problem aims to predict a sequence of action steps that can drive the current visual state toward the goal visual state. The authors of this paper argue that visual observations and the reasoning process are fixed (i.e., kept constant) throughout the planning process, which fails to account for the interactivity of agents and uncertainty that typically arises in the real world.

The paper proposes active procedure planning in instructional videos; in this setting, agents (e.g., a planning model) can make additional intermediate observations to disambiguate the states and constrain the planning trajectories during testing. The central challenge then becomes how to select a subset of intermediate additional observations to enhance planning accuracy while keeping the cost within a predefined budget. The proposed approach is uncertainty-aware, meaning an agent will use additional observations only when there is uncertainty. This is achieved by measuring the uncertainty arising from task sequence variation and the uncertainty related to the model's prediction confidence.

Experimental results provide empirical evidence regarding the effect of active procedure planning with uncertainty awareness on the accuracy of the generated plans.

**Strengths:**

The paper introduces active procedure planning, a novel problem setup that provides a unique lens on the task of procedure planning in instructional videos. The motivations and ideas behind this paper are sound and convincing. Additionally, the proposed uncertainty-aware approach is innovative.

The contributions of this paper are fairly substantial. For instance, the methodology represents a paradigm shift in procedure planning in instructional videos, transitioning from passive reasoning to a more dynamic and adaptive form of learning and reasoning.

Furthermore, the paper is well-written, providing an enjoyable and enlightening read.

**Weaknesses:**

Some methodology and technical details are not clear, and some analyses are missing (e.g., efficiency-efficacy trade-off); see Questions below.

**Questions:**

1. How have you incorporated the active planning mechanism into the diffusion-based procedure planning models? That is, how does the inference model depicted in Figure 2 accommodate the additional set S? The model architecture was not originally designed or trained to accept additional visual observations. What modifications did you implement for these models? Are the adjustments you made easily transferable to non-diffusion model-based procedure planning models?

2. While additional visual observations lead to improvements in the generated procedure plans, they also increase the computational burden. There is a clear efficiency-performance tradeoff, yet an analysis of this is currently absent. Could you quantify the increase in inference time and memory usage incurred while achieving performance gains?

3. From an application standpoint, how realistic or significant is the proposed active procedure planning setup? How could additional intermediate visual observations be made available at the time of testing? Conventional procedure planning is understandable, as one can envision a scenario where a human provides a robot with a photo of available ingredients (initial visual observation) and a photo of a prepared dish (goal visual observation), asking the robot to generate a sequence of steps as the procedure plan. However, it is challenging to imagine how a robot could access intermediate additional visual observations when it is uncertain in the aforementioned application scenario.

---

> ### Author Response · Authors · 2023-11-16
>
> Thank you for the insightful comments!
> ## Q1: Implementation of diffusion model and alternative models
>
> In our implementation, we trained two diffusion models, one without additional observations (VOs)(model A) and one with VOs (model B). In the diffusion model, the tensor that holds the VOs is fixed as “0” in model A; and it is set as the embedding of the VO in model B. During inference time, when an instance is selected to include additional observations, we use model B to perform the inference; otherwise, we use model A.
>
> So far, we implemented our method based on a diffusion model but the concept is appliable to other model structures. There are two prevalent models for weakly-supervised procedure planning. Other than PDPP (diffusion-based), the other is P3IV [1], which combines transformer and GAN. However, P3IV codes are not published and the paper lacks adequate technical details to be re-implemented. Other legacy procedure planning methods adopted supervised learning, which assumed access to all intermediate observations. This is incompatible with the assumptions of active procedure planning, making them not comparable. Therefore, they cannot be tested in the current problem context.
>
> [1] Henghui Zhao, Isma Hadji, Nikita Dvornik, Konstantinos G. Derpanis, Richard P. Wildes, and Al- lan D. Jepson. P3IV: Probabilistic procedure planning from instructional videos with weak super- vision. In CVPR, 2928–2938, 2022
>
> ## Q2: Efficiency-performance tradeoff
>
> As mentioned earlier, we trained two models: Model A that uses only start and goal observations, and Model B that incorporates additional observations. During inference, we initially employ Model A to make predictions. If the uncertainty score of the action prediction exceeds a predefined threshold, we switch to Model B to predict actions. Based on the above design, active procedure planning involves two computational overheads as explained below.
>
> * Training two models (A & B) rather than one (i.e. only Model A is needed in passive procedure planning), which doubles the training time. In our implementation, a model with PDPP framework can be trained in about 3 hours on CrossTask dataset and 5.5 hours on COIN based on the current hardware configuration (4 NVIDIA RTX A5000 GPUs; Please refer to Implementation Details for more information on training configurations). Training time on MDPP framework is about 8 hours on CrossTask and 12 hours on COIN. Note that the time and memory needed for training both models A and B are identical under the same conditions. Since training happens only once, it is affordable in the current problem.
>
> * During inference phase, a subset of instances will be selected to perform additional observations (due to high uncertainty score). They will go through both Models A and B, instead of only Model A in passive procedure planning. The inference time on individual models is identical (so is the memory usage). Therefore, the computational time for those selected instances is doubled. The total inference time overhead is dependent on the proportion of instances that require additional observation, which is in turn contingent on the uncertainty threshold. For example, if at a given threshold, 30% of instances require additional observation, the inference time of active procedure planning will be 30% higher than passive planning.
>
> We will add the above explanations in the revised manuscript.
>
> ## Q3: Application scenarios
>
> In this study, we focus on an initial investigation of methodological issues in active procedure planning and evaluate it on an established protocol and datasets, i.e. procedure planning in instructional videos. In the example where a robot is tasked to make a plan (thanks for the reviewer’s good suggestion), a robot may need to observe the current scene and come up with a plan. When the robot has low confidence in the prediction, it would be costly or even dangerous to execute it blindly. The major difference between the above robotics scenario and procedure planning in instructional videos is that for the latter case, the intermedia states are readily available, while in the former case, the robot needs to take certain actions (e.g., navigate to an alternative perspective, move an object to reduce occlusion, ask human for intervention, etc.) to actuate new observations. Although these actions may pose extra challenges beyond procedure planning on instructional videos, we believe that the basic concept of leveraging uncertainty estimation and incorporating new observation information in the decision loop is generic and re-usable in other situations.
>
> In summary, the scope of the current paper is to study the effect of active planning in the context of instructional videos. We put it as future work to implement alternative practical scenarios (e.g., a robot makes a plan, executes it, and observes the effects), which may lead to many more impactful outcomes.

---

> > ### Comment · Reviewer_9LnP · 2023-11-21
> >
> > I appreciate the authors' response! Overall, I find the ideas presented in this paper relatively new and interesting. However, for the revised manuscript, I would like to suggest that the authors consider making the following clarifications (please feel free to correct me if I have misunderstood anything):
> >
> > 1. The proposed setting, i.e., active procedure planning, requires **annotations** of intermediate observations during both the training and testing phases.
> > 2. The analysis of the efficiency-performance tradeoff would differ when considering diffusion model-based architectures like PDPP or MPDPP compared to other frameworks such as transformer-based architectures. In the latter case, the integration of additional visual observations may require extra tokens, which is not the case with diffusion-based models, where zero padding dimensions are replaced with features of these additional observations.
> >
> > Additionally, I would like to bring to the authors' attention that there are two recent ICCV 2023 papers on procedure planning in instructional videos [1-2]. Notably, both of these papers do not require annotations of intermediate observations. Therefore, I recommend that the authors discuss these papers in their revision and consider applying their proposed method to these non-diffusion model-based architectures.
> >
> > [1] Li, Zhiheng, Wenjia Geng, Muheng Li, Lei Chen, Yansong Tang, Jiwen Lu, and Jie Zhou. "Skip-Plan: Procedure Planning in Instructional Videos via Condensed Action Space Learning." In Proceedings of the IEEE/CVF International Conference on Computer Vision, pp. 10297-10306. 2023.
> >
> > [2] Wang, An-Lan, Kun-Yu Lin, Jia-Run Du, Jingke Meng, and Wei-Shi Zheng. "Event-Guided Procedure Planning from Instructional Videos with Text Supervision." In Proceedings of the IEEE/CVF International Conference on Computer Vision, pp. 13565-13575. 2023.
> >
> > Furthermore, the P3IV GitHub repository: https://github.com/SamsungLabs/procedure-planning
> >
> > I understand that these recent papers and the GitHub repository mentioned may not have been available at the time of preparing this submission.
> >
> > In conclusion, my positive opinion of this paper remains unchanged due to the strengths I have mentioned. I will maintain my rating accordingly.

---

> > > ### Author Response · Authors · 2023-11-22
> > > **We appreciate the reviewer's comments and the decision to maintain a positive rating!**
> > >
> > > * We affirm the two points raised, i.e., additional annotation on intermediate observation, and extra computation load on transformer-based models (as different from diffusion-based). We will clarify these in the revised manuscript.
> > >
> > > * Thanks for directing us to the latest works and updated repository of P3IV! As suggested by the reviewer, we will discuss them in the revised manuscript; and explore how to implement them as benchmarks in near future.

---

### Meta-Review · Area_Chair_zcZd · 2023-12-09

**Metareview:**

The paper introduces a novel approach to active procedure planning in instructional videos, where agents can make intermediate observations to improve planning accuracy, addressing the real-world uncertainty and interactive nature of tasks. The empirical results demonstrate the efficacy of this approach in enhancing the accuracy of the generated plans in certain applications.

The discussions reveal a mix of strengths and areas for improvement: The paper introduces an innovative concept of active procedure planning in instructional videos; this novel approach represents a methodological shift towards more dynamic and interactive learning and reasoning in procedure planning. The presentation is also commended for the engaging narrative among the reviewers.

Key areas of improvements include the need for greater clarity in the paper's methodology, particularly how the active planning mechanism integrates with existing models. There is also skepticism about the practical applicability of the proposed method in real-world scenarios, as well as further clarification on the computational resources and time implications of the proposed method. Additionally, the paper’s approach to handling large action spaces and scenarios with unseen event combinations requires further exploration and validation.

**Justification For Why Not Higher Score:**

While the innovative nature of the paper and its potential impact are clear, the lack of detailed technical clarity, especially in method integration and practical applicability, limits its current standing. The absence of a thorough analysis of the efficiency-performance tradeoff and the handling of complex scenarios also indicate areas needing further development to enhance the significance of the contribution.

**Justification For Why Not Lower Score:**

All reviewers acknowledged the novelty of the problem setting, and the presentation is clear and well-received. The authors have shown enthusiastic engagement in addressing the reviewers' concerns.

---

### Decision · Program_Chairs · 2024-01-16

Reject